# Implementation and Performance Evaluation of the Frequency-Domain-Based Bit Flipping Controller for Stabilizing the Single-Bit High-Order Interpolative Sigma Delta Modulators

**Huishan Zhai and Bingo Wing-Kuen Ling ***

School of Information Engineering, Guangdong University of Technology, Guangzhou 510006, China; hs617807525@163.com
* Correspondence: yongquanling@gdut.edu.cn; Tel.: +86-203-932-2438; Fax: +86-203-932-2252

**Abstract:** This paper is an extension of the existing works on the frequency-domain-based bit flipping control strategy for stabilizing the single-bit high-order interpolative sigma delta modulator. In particular, this paper proposes the implementation and performs the performance evaluation of the control strategy. For the implementation, a frequency detector is used to detect the resonance frequencies of the input sequence of the sigma delta modulator. Then, a neural-network-based controller is used for finding the solution of the integer programming problem. Finally, the buffers and the combinational logic gates as well as an inverter are used for implementing the proposed control strategy. For the performance evaluation, the stability region in terms of the input dynamical range is evaluated. It is found that the control strategy can significantly increase the input dynamical range from 0.24 to 0.58. Besides, the control strategy can be applied to a wider class of the input signals compared to the clipping method.

**Keywords:** high-order interpolative sigma delta modulator; bit flipping control; quantization; fractal; chaos; complex number

## 1. Introduction

A single-bit high-order interpolative sigma delta modulator [1,2] consists of a negative feedback of a high order loop filter and a single bit quantizer [3]. If the loop filter is designed properly, then the magnitude of the noise transfer function can be very small at the signal band and the noise is mainly localized outside the signal band. On the other hand, the magnitude of the signal transfer function can also be very small outside the signal band and the signal is mainly localized in the signal band. As the signal and the noise are separated in different frequency bands, a very good analog to digital conversion performance can be achieved by applying a simple lowpass filtering at the quantizer output. This technique is known as the noise shaping technique [1,4]. As an oversampling operation for an audio signal can be implemented using an existing hardware [5,6], a single-bit high-order interpolative sigma delta modulator is widely employed for the analog to digital conversion in the audio devices [7,8].

However, the loop filter is required to be unstable (not bounded input bounded output stable) in order to achieve a high signal to noise ratio. Therefore, a single-bit high-order interpolative sigma delta modulator may suffer from the internal instability problem [9,10]. That is, the state variables of the loop filter may diverge. To address this issue, control is required for the stabilization purpose [7,11–13].

The most common control method is clipping. Here, the values of the output of the loop filter are clipped to a certain value when they exceed a certain value [7]. However, this control method

usually results to the occurrence of a limit cycle [14]. This is because it will reach the same set of state vectors every time when the control action takes place. Another common control method is to reset the state vector to a certain vector in its largest invariant set if the largest invariant set of the state vectors exists [11]. However, the determination of the largest invariant set of the state vectors requires a very high computational power [15]. Therefore, the implementation cost of this approach is very expensive.

To address the drawbacks of the existing clipping control strategy, a sliding mode control-based method was proposed [12,13]. However, the conventional sliding mode control technique usually requires an amplifier with the gain higher than the saturation level of the quantizer. Equivalently speaking, this flipping strategy is valid only for the input signal with a very small dynamic range.

To address the drawbacks of the above control strategies, a frequency-domain-based bit flipping control strategy was proposed [1]. Although the theoretical analysis was discussed in detail in [1], the implementation of this control strategy and the detail performance evaluation have not performed. The main contribution of this paper is to propose an implementation to realize this control strategy such that the stability region in terms of the input dynamical range can increase significantly.

The outline of this paper is as follows. In Section 2, the implementation of the frequency-domain-based bit flipping controller is proposed. In Section 3, the performance evaluations are presented. Finally, a conclusion is drawn in Section 4.

## 2. Implementation of the Frequency-Domain-Based Bit Flipping Controller

Since this control strategy is based on the cancellation of certain frequency contents of the input sequence of the sigma delta modulator, it only requires a frequency detector [16–18] to detect the resonance frequencies of the input sequence of the sigma delta modulator, a neural-network-based controller for finding the solution of the integer programming problem [19–21], the buffers and the combinational logic gates as well as an inverter for implementing the proposed control strategy. The block diagram of the proposed method is shown in Figure 1.

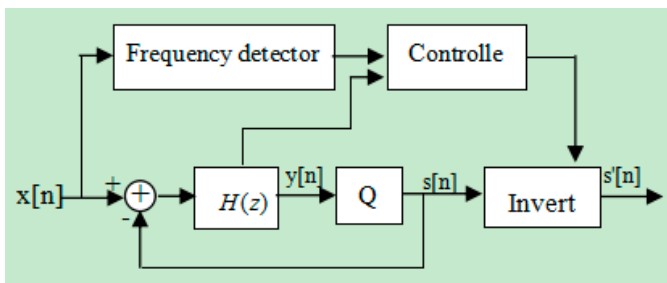

**Figure 1.** Block diagram of the proposed system.

First, the frequency detector detects the existence of the resonance frequencies in the input sequence. Here, the resonance frequencies relate to those eigenvalues of **A** lying on the unit circle. To detect the resonance frequencies, it is equivalent to test whether $\sum_{n \geq 0} u[n]d_i^{-n}$ is bounded or not with $|d_i| = 1$ for those $i \in \{0, \cdots, P-1\}$. However, as the length of the signal is infinite, the implementation involves the infinite sum. This has an implementation difficulty. To address this issue, a buffer is employed to preload a block of signal and the summation is evaluated only over the points in the block. Now, a simple accumulator is employed for computing the summation of the real part of the points in the block. Similarly, another simple accumulator is employed for computing the summation of the imaginary part of the points in the block. Finally, two comparators are employed to test whether both the absolute values of the summed real part and the summed imaginary part of the points in the block are larger than a threshold value or not.

Second, the loop filter is implemented via the direct form structure. Here, only the delay cells, the multipliers and the adders are required for the implementation. The state variables of the digital filter are stored in the delay cells.

Third, the quantizer is implemented via the comparators. The output of the loop filter is compared to the quantization levels of the quantizer via the comparators. Finally, the quantizer gives the output which is the output of the corresponding comparator.

Fourth, the controller implements the integer programming algorithm via a neural network. The decision vector of the optimization problem is the weight vector of the neural network. The update of the decision vector is implemented via the update of the weight vector of the neural network. Once the update of the weight vector is converged, the converged weight vector is used to control the inverter.

Finally, the bit streams of the output of the quantizer is compared to the converged weight vector of the neural network. If the bit streams of the output of the quantizer are different from the converged weight vector of the neural network, then the corresponding bit is inverted by the inverter. Otherwise, the bit streams of the output of the quantizer remain unchanged.

## 3. Performance Evaluation

Since the control objective is to guarantee the stability of the sigma delta modulator and the stability only depends on the input signal for a given sigma delta modulator and the control strategy, the input dynamical range is employed as the criterion for the comparison.

As discussed in Section 1 that the clipping method [7] is the most common control method, the clipping method [7] is compared. More precisely, the clipping rule is to reset the state vector $\mathbf{x}(k)$ to the zero vector when the absolute values of the outputs of the loop filter are greater than or equal to one.

In this paper, a single bit fifth order interpolative sigma delta modulator realized using the state space representation with the following state space matrices is considered:

$$\mathbf{A} = \begin{bmatrix} 0.9990 & 1.0000 & 0.0000 & 0.0000 & 0.0000 \\ -0.0020 & 0.9990 & 0.0530 & 0.1764 & 0.5164 \\ 0 & 0 & 0.9997 & 1.0000 & -0.0000 \\ 0 & 0 & -0.0007 & 0.9997 & 0.3567 \\ 0 & 0 & 0 & 0 & 1.0000 \end{bmatrix},$$

$$\mathbf{B} = \begin{bmatrix} 0 & 0 & 0 & 0 & 1 \end{bmatrix}^T$$

and

$$\mathbf{C} = \begin{bmatrix} 0.0307 & 0.4153 & 0.0828 & 0.2754 & 0.8063 \end{bmatrix}$$

This sigma delta modulator is chosen for the comparison because it is widely used in the audio industry [7].

It is worth noting that the eigenvalues of $\mathbf{A}$ are $0.9990 + 0.0445j$, $0.9990 - 0.0445j$, $0.9997 + 0.0264j$, $0.9990 - 0.0264j$ and 1. As all the eigenvalues are on the unit circle, this loop filter is not bounded input bounded output stable. For the ease of the implementation, assume that the initial state vector is zero. Since this loop filter contains a DC pole, the step input is applied to this sigma delta modulator for illustration purposes. This is because this input will result to the occurrence of the resonance. Therefore, control is required for stabilizing the sigma delta modulator. Here, three-step input signals with different step sizes are illustrated. For the first case, an input step size equal to 0.24 is applied to the sigma delta modulator. That is, $u(k) = 0.24$ for $k \geq 0$. Figure 2a,b show the outputs of the loop filters when our control strategy and the clipping method [7] are applied, respectively. Figure 2c,d show the magnitude responses of the outputs of the loop filters when our control strategy and the clipping method [7] are applied, respectively. It can be seen from Figure 2a that the output of the loop filter under our proposed control strategy is bounded. This implies that there exists a binary sequence in $Y_{d_i}$ or the neural network finds a solution of Problem (P) such that the objective functional value of Problem (P) is exactly equal to zero. Besides, it can be seen from Figure 2c that the output of the

loop filter is a wide band signal. Its frequency content is very rich. This means that the sigma delta modulator exhibits the chaotic behavior. On the other hand, it can be seen from Figure 2b that the output of the loop filter under the clipping strategy [7] is periodic. This is because the state vectors are reset to the same state every time when the reset action takes place. Besides, it can be seen from Figure 2d that the output of the loop filter only consists of the DC frequency and the Nyquist sampling frequency. This implies that the sigma delta modulator exhibits the limit cycle behavior. For the audio application, this frequency component with the high amplitude refers to the annoying audio tune, which should be avoided.

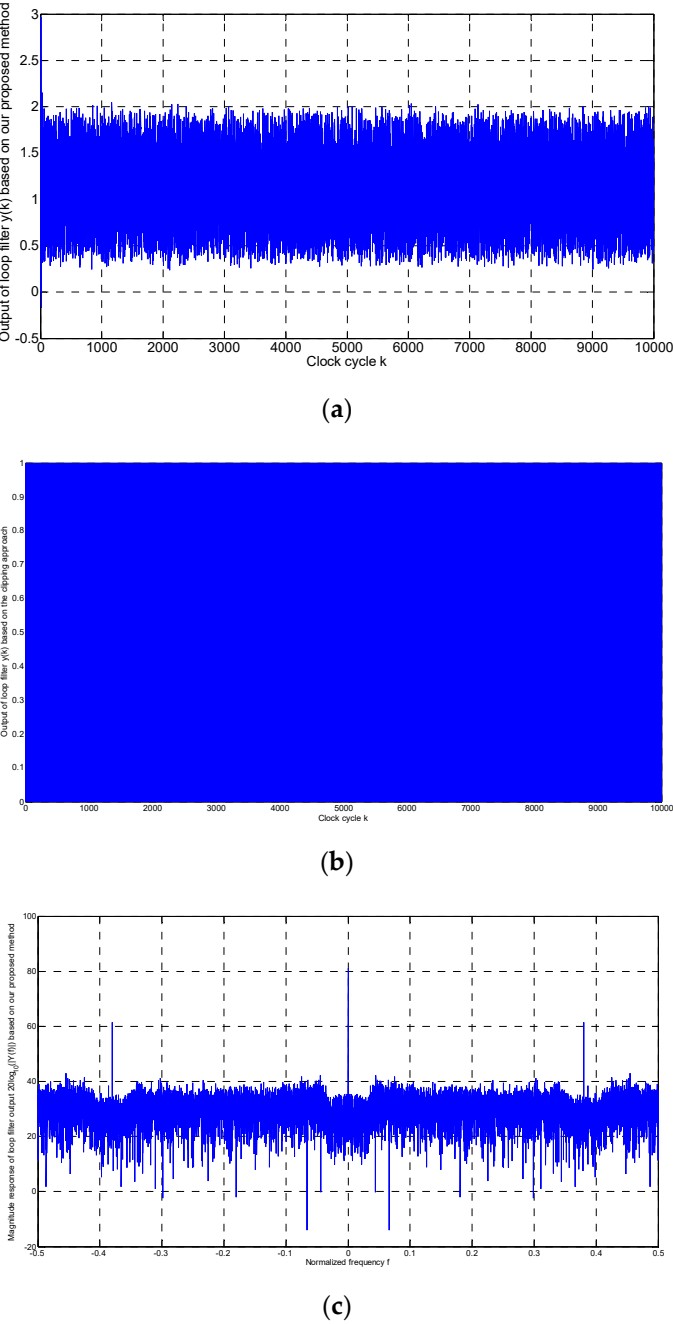

(a)

(b)

(c)

**Figure 2.** *Cont.*

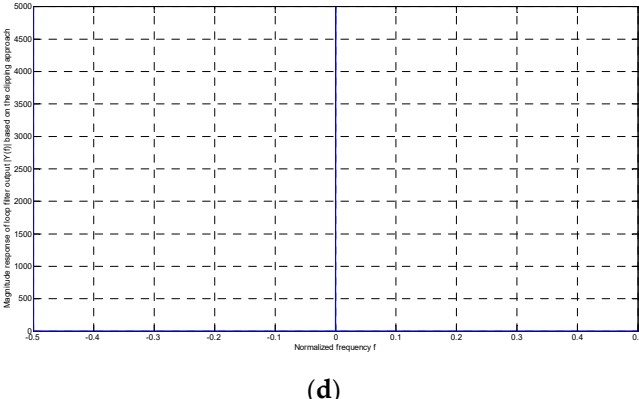

**(d)**

**Figure 2.** (**a**) Output of the loop filter when $u(k) = 0.24$ for $k \geq 0$ under our proposed control strategy. (**b**) Output of the loop filter when $u(k) = 0.24$ for $k \geq 0$ under the clipping strategy [7]. (**c**) Magnitude response of the output of the loop filter when $u(k) = 0.24$ for $k \geq 0$ under our proposed control strategy. (**d**) Magnitude response of the output of the loop filter when $u(k) = 0.24$ for $k \geq 0$ under the clipping strategy [7].

In order to evaluate the performance on the analog to digital conversion of the sigma delta modulator under various control strategies, the peak signal to noise ratio is employed as the performance index. Since the passband of the loop filter is $\left(-\frac{\pi}{64}, \frac{\pi}{64}\right)$, the ideal lowpass filter with the passband equal to $\left(-\frac{\pi}{64}, \frac{\pi}{64}\right)$ is applied to the quantizer output. Here, the ideal lowpass filtering is implemented using the discrete Fourier transform approach. That is, those discrete Fourier transform coefficients outside the frequency band $\left(-\frac{\pi}{64}, \frac{\pi}{64}\right)$ are set to zero. It is found that the signal to noise ratio of the signal based on our control strategy is 36.98 dB. On the other hand, the signal to noise ratio of the signal based on the clipping method [7] is $-1.76$ dB. This is because the output of the loop filter only consists of the DC frequency and the Nyquist sampling frequency while there is no DC component in the input signal. Therefore, the signal to noise ratio of the signal is significantly worse.

Now, consider another input step size. Here, the input step size is incremented by 0.01. More precisely, an input step size of 0.25 is applied. That is, $u(k) = 0.25$ for $k \geq 0$. Figure 3a,b show the outputs of the loop filters when our control strategy and the clipping method [7] are applied, respectively. Figure 3c,d show the magnitude responses of the outputs of the loop filters when our control strategy and the clipping method [7] are applied, respectively. It can be seen from Figure 3a,c that the sigma delta modulator exhibits the similar behavior as before if our proposed control strategy is applied. On the other hand, it is worth noting that the increase in the input step size causes the absolute value of the output of the loop filter greater than or equal to one at every time instant if no control force is applied. Therefore, if the clipping control strategy [7] is applied, then the clipping action takes place at every time instant. Hence, it can be seen from Figure 3b,d that the output of the loop filter is always equal to zero if the clipping control strategy [7] is applied. In this case, the sigma delta modulator is not working properly.

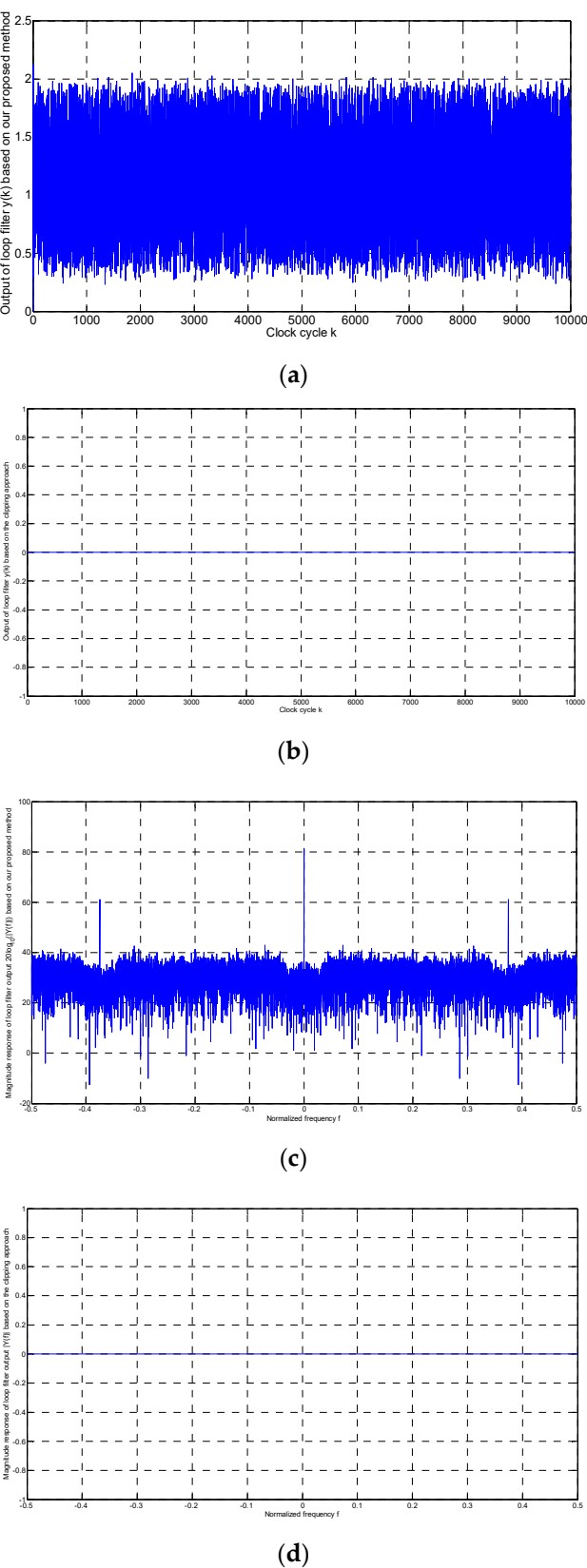

**Figure 3.** (**a**) Output of the loop filter when $u(k) = 0.25$ for $k \geq 0$ under our proposed control strategy. (**b**) Output of the loop filter when $u(k) = 0.25$ for $k \geq 0$ under the clipping strategy [7]. (**c**) Magnitude response of the output of the loop filter when $u(k) = 0.25$ for $k \geq 0$ under our proposed control strategy. (**d**) Magnitude response of the output of the loop filter when $u(k) = 0.25$ for $k \geq 0$ under the clipping strategy [7].

Likewise, it is found that the signal to noise ratio of the signal based on our control strategy is 35.37 dB. On the other hand, the signal to noise ratio of the signal based on the clipping method [7] is exactly 0 dB.

Next, a very large input step, which is equal to 0.58, is applied. Figure 4a,b show the outputs of the loop filters when our control strategy and the clipping method [7] are applied, respectively. Figure 4c,d show the magnitude responses of the outputs of the loop filters when our control strategy and the clipping method [7] are applied, respectively. It can be seen from Figure 4a,c that the sigma delta modulator also exhibits the similar behavior as before if our proposed control strategy is applied. Similarly, it can be seen from Figure 4b,d that the output of the loop filter is always zero under the clipping control strategy [7]. From here, it can be seen that our proposed control method can achieve a larger stability region in terms of the input dynamical range.

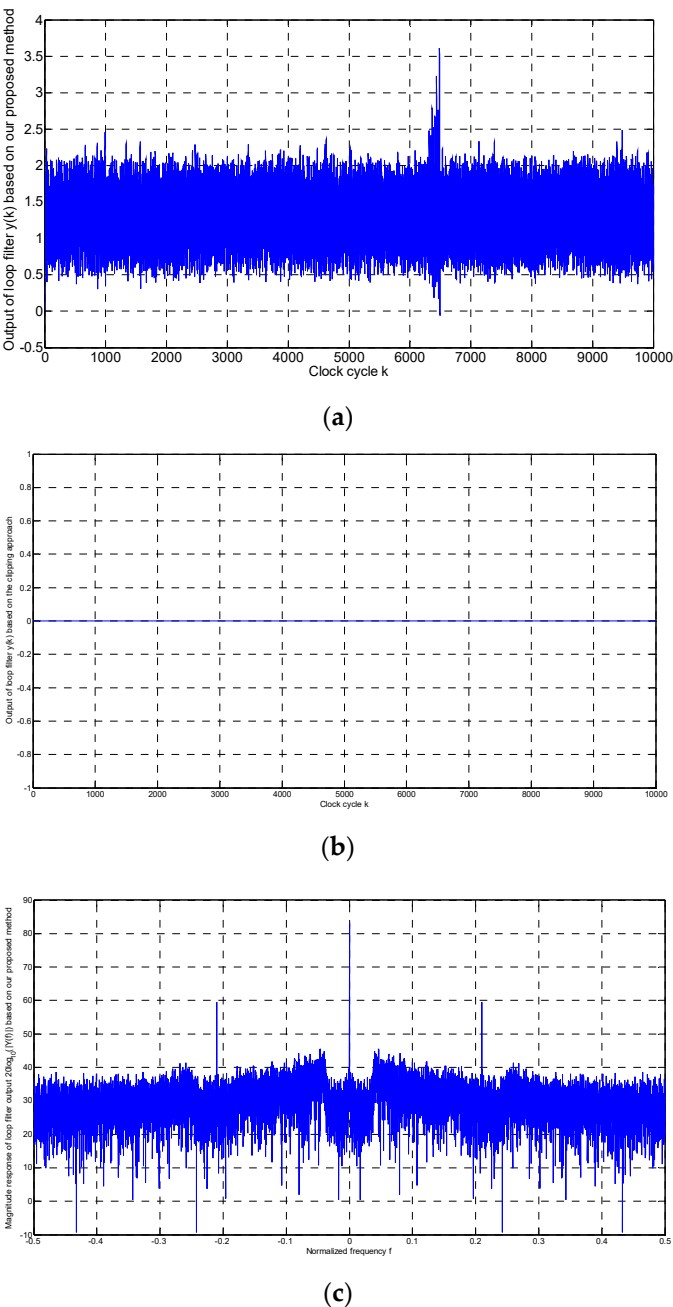

(a)

(b)

(c)

**Figure 4.** *Cont.*

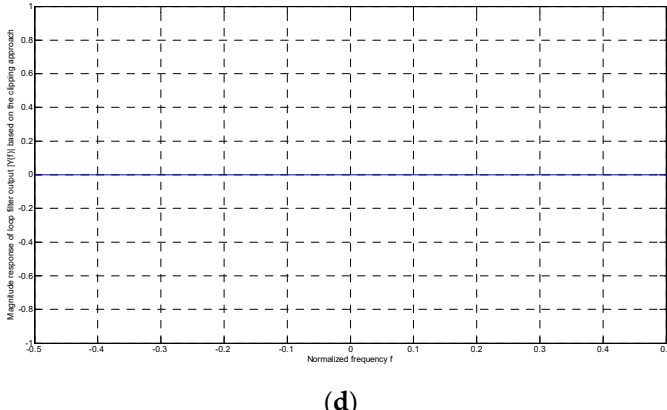

**(d)**

**Figure 4.** (**a**) Output of the loop filter when $u(k) = 0.58$ for $k \geq 0$ under our proposed control strategy. (**b**) Output of the loop filter when $u(k) = 0.58$ for $k \geq 0$ under the clipping strategy [7]. (**c**) Magnitude response of the output of the loop filter when $u(k) = 0.58$ for $k \geq 0$ under our proposed control strategy. (**d**) Magnitude response of the output of the loop filter when $u(k) = 0.58$ for $k \geq 0$ under the clipping strategy [7].

Likewise, it is found that the signal to noise ratio of the signal based on our control strategy is 32.52 dB. On the other hand, the signal to noise ratio of the signal based on the clipping method [7] is exactly 0 dB.

To further demonstrate the effects of the input step size on the stability of the sigma delta modulator, Figure 5a,b show the maximum absolute value of the output of the loop filter against the input step size under our proposed control strategy and the clipping control strategy [7], respectively. For our proposed control strategy, it can be seen from Figure 5a that the input step sizes being smaller than or equal to 0.58 will yield the maximum absolute values of the outputs of the loop filter being equal to nonzero. This means that the sigma delta modulator can be stabilized. However, the modulus of $U(z)\big|_{z=d_i}$ increases as the input step size increases. Eventually, $\frac{d_i \widetilde{x}_i(0)}{\widetilde{b}_i} + U(z)\big|_{z=d_i} \notin Y_{d_i}$. In this case, the sigma delta modulator cannot be stabilized. Therefore, it can be seen from Figure 5a that the maximum absolute values of the outputs of the loop filter are equal to zero when the input step sizes are larger than 0.58. On the other hand, for the clipping control method [7], it can be seen from Figure 5b that the input step sizes being smaller than or equal to 0.24 will yield the maximum absolute values of the outputs of the loop filter being equal to nonzero. However, the maximum absolute values of the outputs of the loop filter are equal to zero when the input step sizes are larger than 0.58. This also demonstrates that our proposed control method yields a wider stability region in terms of the input dynamical range.

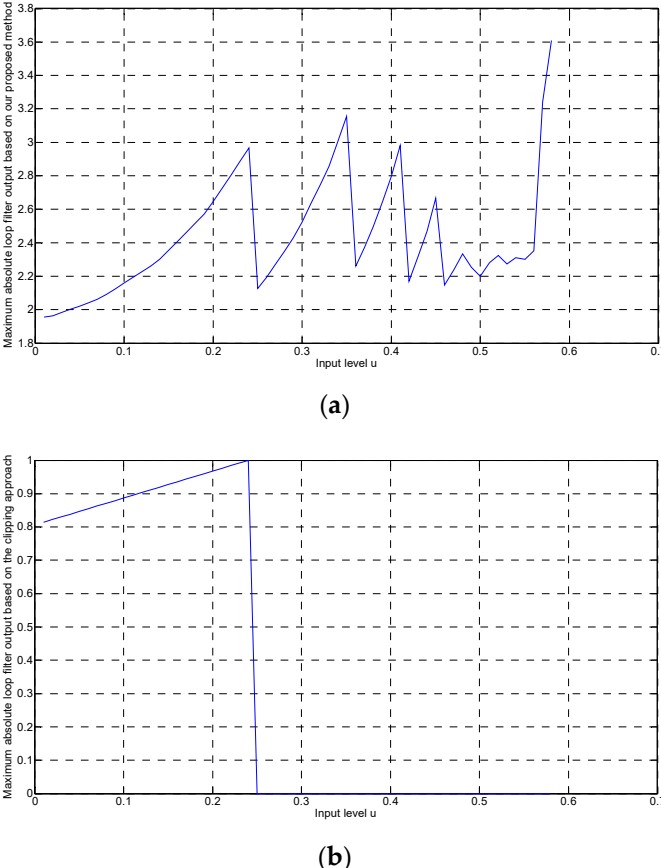

**Figure 5.** (**a**) The maximum absolute value of the output of the loop filter against the input step size under our proposed control strategy. (**b**) The maximum absolute value of the output of the loop filter against the input step size under the clipping control strategy [7].

As the phases of the eigenvalues of $A$ are $0.0142\pi$, $-0.0142\pi$, $0.0084\pi$, $-0.0084\pi$ and $0$, the sinusoidal inputs with the angular frequencies equal to $0.0142\pi$ and $0.0084\pi$ are applied for the illustration purpose. Here, the amplitude of the sinusoidal input is equal to $0.24$, which is chosen the same as before for the comparison purpose. For the simplicity reason, there is no phase shift on the sinusoidal input. Figure 6a,b show the outputs of the loop filters when the angular frequency is equal to $0.0142\pi$ under our control strategy and the clipping method [7], respectively. Figure 6c,d show the magnitude responses of the outputs of the loop filters when the angular frequency is equal to $0.0142\pi$ under our control strategy and the clipping method [7], respectively. Figure 7a,b show the outputs of the loop filters when the angular frequency is equal to $0.0084\pi$ under our control strategy and the clipping method [7], respectively. Figure 7c,d show the magnitude responses of the outputs of the loop filters when the angular frequency is equal to $0.0084\pi$ under our control strategy and the clipping method [7], respectively. It can be seen from both Figures 6 and 7 that the sigma delta modulator exhibits the similar behavior as before for our proposed control strategy.

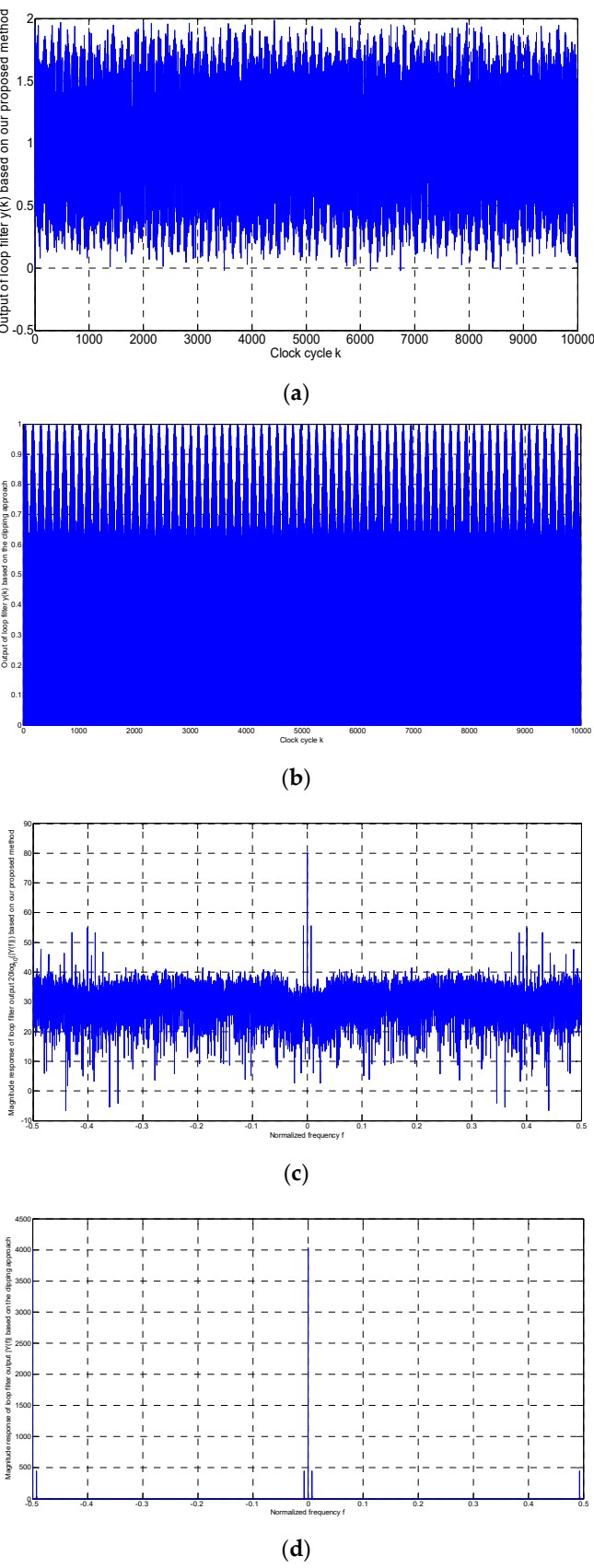

**Figure 6.** (**a**) Output of the loop filter when $u(k) = 0.24\sin(0.0142\pi k)$ for $k \geq 0$ under our proposed control strategy. (**b**) Output of the loop filter when $u(k) = 0.24\sin(0.0142\pi k)$ for $k \geq 0$ under the clipping control strategy [7]. (**c**) Magnitude response of the output of the loop filter when $u(k) = 0.24\sin(0.0142\pi k)$ for $k \geq 0$ under our proposed control strategy. (**d**) Magnitude response of the output of the loop filter when $u(k) = 0.24\sin(0.0142\pi k)$ for $k \geq 0$ under the clipping control strategy [7].

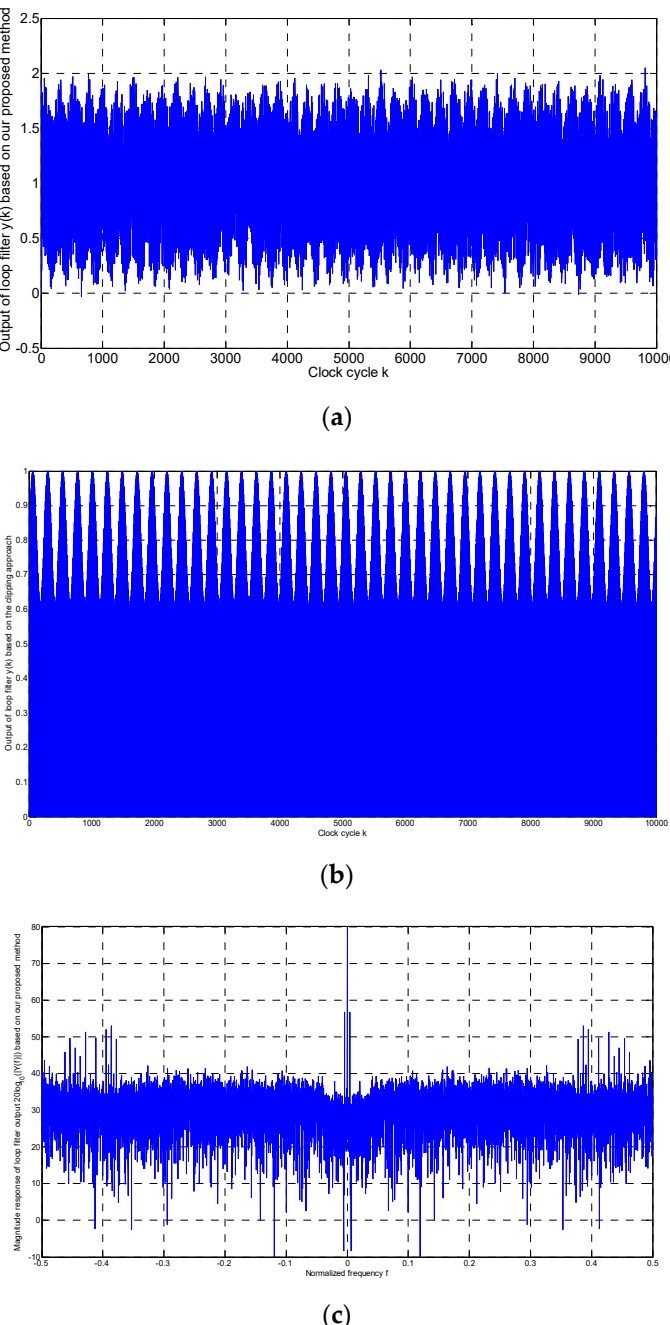

**Figure 7.** *Cont.*

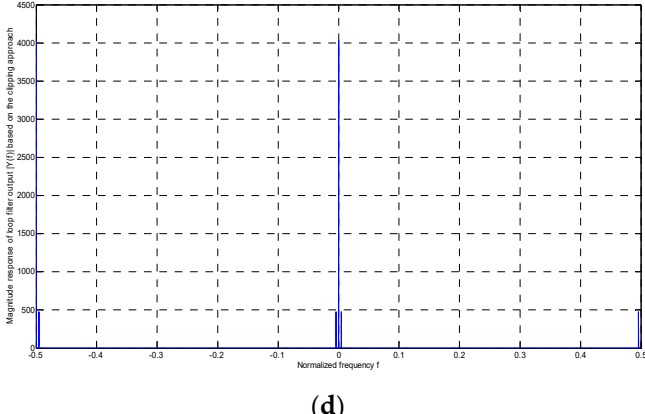

(**d**)

**Figure 7.** (**a**) Output of the loop filter when $u(k) = 0.24\sin(0.0084\pi k)$ for $k \geq 0$ under our proposed control strategy. (**b**) Output of the loop filter when $u(k) = 0.24\sin(0.0084\pi k)$ for $k \geq 0$ under the clipping control strategy [7]. (**c**) Magnitude response of the output of the loop filter when $u(k) = 0.24\sin(0.0084\pi k)$ for $k \geq 0$ under our proposed control strategy. (**d**) Magnitude response of the output of the loop filter when $u(k) = 0.24\sin(0.0084\pi k)$ for $k \geq 0$ under the clipping control strategy [7].

To demonstrate the application value of our proposed method, the sigma delta modulator is applied to an electromyogram [1]. Figure 8 shows the electromyogram. Figure 9a,b show the outputs of the loop filters under our control strategy and the clipping method [7], respectively. Figure 9c,d show the magnitude responses of the outputs of the loop filters under our control strategy and the clipping method [7], respectively. It can be seen from Figure 9a,c that the sigma delta modulator exhibits similar behavior to before for our proposed control strategy. This demonstrates the generality of our proposed method and the possibility of applying our proposed method to some practical signals.

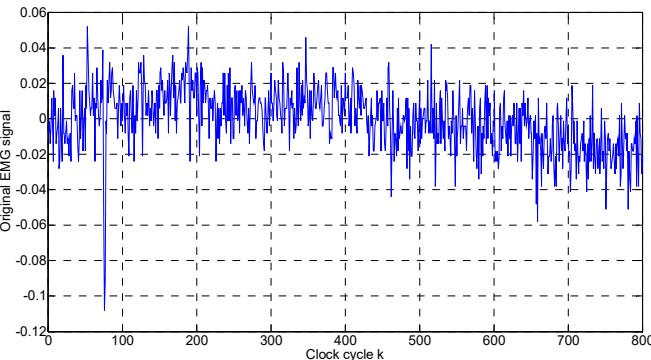

**Figure 8.** Electromyogram.

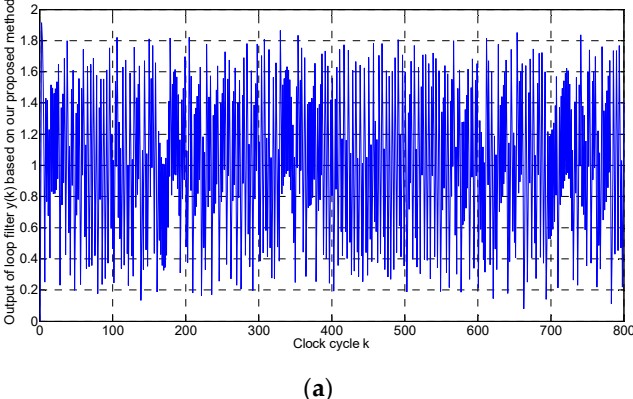

(**a**)

**Figure 9.** *Cont.*

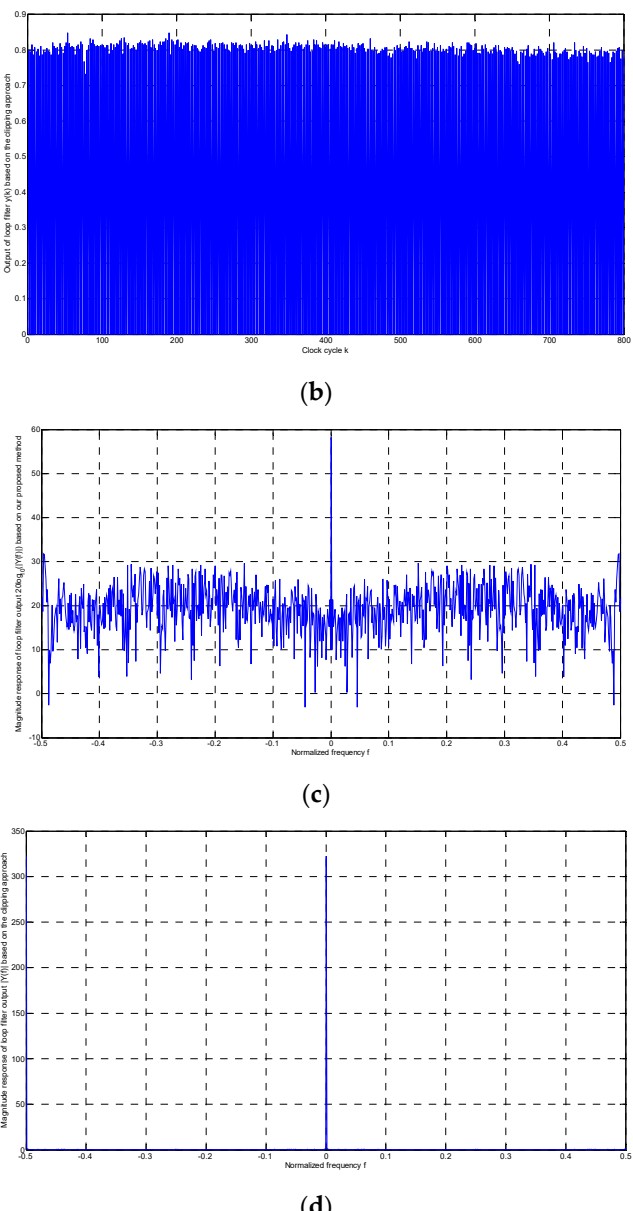

**Figure 9.** (**a**) Output of the loop filter under our proposed control strategy when the input is an electromyogram. (**b**) Output of the loop filter under the clipping control strategy [7] when the input is an electromyogram. (**c**) Magnitude response of the output of the loop filter under our proposed control strategy when the input is an electromyogram. (**d**) Magnitude response of the output of the loop filter under the clipping control strategy [7] when the input is an electromyogram.

## 4. Conclusions

This paper proposed an implementation of the frequency-domain-based bit flipping control strategy [1] for stabilizing the single-bit high-order interpolative sigma delta modulator. In particular, a frequency detector was used to detect the resonance frequencies of the input sequence of the sigma delta modulator. Moreover, a neural-network-based controller was used for finding the solution of the integer programming problem. Moreover, the buffers and the combinational logic gates as well as an inverter were used for implementing the control strategy. The implementation cost was low. The computer numerical simulation results show that the stability region in terms of the input dynamical range increased from 0.24 to 0.58. Moreover, the control strategy could be applied to a wider class of the input signals compared to the clipping method.

**Author Contributions:** Conceptualization, H.Z. and B.W.-K.L.; methodology, software, H.Z. and B.W.-K.L.; validation, B.W.-K.L.; formal analysis, H.Z. and B.W.-K.L.; investigation, H.Z. and B.W.-K.L.; resources, B.W.-K.L.; data curation, H.Z.; writing—original draft preparation, H.Z.; writing—review and editing, H.Z.; visualization, H.Z.; supervision, B.W.-K.L.; project administration, B.W.-K.L.; funding acquisition, B.W.-K.L. All authors have read and agreed to the published version of the manuscript.

**Funding:** This research was funded partly by the National Nature Science Foundation of China [no. U1701266, no. 61372173 and no. 61671163], the Team Project of the Education Ministry of the Guangdong Province [no. 2017KCXTD011], the Guangdong Higher Education Engineering Technology Research Center for Big Data on Manufacturing Knowledge Patent [no. 501130144], the Guangdong Province Intellectual Property Key Laboratory Project [no. 2018B030322016] and Hong Kong Innovation and Technology Commission, Enterprise Support Scheme [no. S/E/070/17].

**Conflicts of Interest:** The authors declare no conflict of interest

## Appendix A. Review on the Conversions between the Integers and the Corresponding Binary Sequences

First, consider the case when a non-negative integer is represented using the octave integer base. Denote mod $(\cdot, \cdot)$ and $\lfloor \cdot \rfloor$ as the modulo operator and the operator taking the smallest integer of a real number, respectively. Define the following nonlinear map: $\Im : Z^+ \cup \{0\} \to \{-1, 1\} \times \{-1, 1\} \times \cdots$, such that $\mathbf{x}' = \Im(n)$ and the $k$th element of $\mathbf{x}'$ is equal to $2\mathrm{mod}\left(\left\lfloor \frac{n}{2^{k-1}} \right\rfloor, 2\right) - 1$. This nonlinear map is to convert a non-negative decimal number $n$ to a vector of a binary sequence $\mathbf{x}'$, in which the elements in $\mathbf{x}'$ are either 1 or $-1$. For example, if $n = 11$, then $\mathbf{x}' = \begin{bmatrix} 1 & 1 & -1 & 1 & -1 & -1 & \cdots \end{bmatrix}^T$. Define a vector $\boldsymbol{\xi}$, in which the $k$th element of $\boldsymbol{\xi}$ is equal to $2^{k-1}$ and denote the vector $\mathbf{1}$ as the vector with all its elements being equal to 1. Then, we have $\boldsymbol{\xi}^T \frac{\mathbf{x}'+1}{2} = n$. This equation is to convert the vector of the binary sequence $\mathbf{x}'$ back to a non-negative decimal number $n$. From here, we can see that $\Im$ is invertible.

For the completeness, consider the case when a non-negative integer is represented using the non-octave integer bases. Similarly, define the following nonlinear map: $\Im' : Z^+ \cup \{0\} \to \{0, \cdots, \lambda' - 1\} \times \{0, 1, \cdots, \lambda' - 1\} \times \cdots$, such that $\mathbf{x}'' = \Im'(n)$ and the $k$th element of $\mathbf{x}''$ is equal to $\mathrm{mod}\left(\left\lfloor \frac{n}{\lambda'^{k-1}} \right\rfloor, \lambda'\right)$. This nonlinear map is to convert a non-negative decimal number $n$ to a vector of an integer sequence $\mathbf{x}''$ in which the elements in $\mathbf{x}''$ are in $\{0, 1, \cdots, \lambda' - 1\}$. Define a vector $\boldsymbol{\xi}'$ in which the $k$th element of $\boldsymbol{\xi}'$ is equal to $\lambda'^{k-1}$. Then, we have $\boldsymbol{\xi}'^T \mathbf{x}'' = n$. This equation is to convert the vector of the integer sequence $\mathbf{x}''$ back to a non-negative decimal number $n$. From here, we can also see that $\Im'$ is invertible.

## Appendix B. Review on Both the Fractal and the Chaotic Behaviors of the Dynamical System with Its State Vectors Being the Corresponding Binary Sequences of the Consecutive Integers

Now, denote $\lambda$ as a complex number with its modulus greater than 1. Define $\Gamma = \left\{\lambda^{-(k-1)} : k \geq 1\right\}$. Let $c_k \in \{1, -1\}$ for $k \geq 1$ be a binary combinational coefficient. Define a vector $\boldsymbol{\psi}$ in which the $k$th element of $\boldsymbol{\psi}$ is $\lambda^{-(k-1)}$. Let $y(n)$ be a complex number that can be represented as a binary combination of the elements in $\Gamma$. That is, $y(n) = \sum_{k \geq 1} c_k \lambda^{-k}$. Denote $Y \equiv \{y(n) : n \geq 0\}$. In this section, we will characterize $Y$. First, the integer 0 is converted to the corresponding binary sequence $\Im(0)$. Then, we can compute the corresponding complex number $y(0)$. Next, increment the integer 0 to the integer 1 and convert the integer 1 to the corresponding binary sequence $\Im(1)$ and compute $y(1)$. Eventually, we obtain $Y$. More precisely, for a given binary sequence $\mathbf{x}(n)$, as $\Im$ is invertible, the corresponding integer is $\boldsymbol{\xi}^T\left(\frac{\mathbf{x}(n)+1}{2}\right)$. Hence, the incremented integer is $1 + \boldsymbol{\xi}^T\left(\frac{\mathbf{x}(n)+1}{2}\right)$ and the corresponding incremented binary sequence is $\Im\left(1 + \boldsymbol{\xi}^T\left(\frac{\mathbf{x}(n)+1}{2}\right)\right)$. As a result, we can define a nonlinear discrete time dynamical system as follows: $\mathbf{x}(n+1) = \Im\left(1 + \boldsymbol{\xi}^T\left(\frac{\mathbf{x}(n)+1}{2}\right)\right)$ and $y(n) = \boldsymbol{\psi}^T\mathbf{x}(n)$ for $n \geq 0$. Here, $\mathbf{x}(n)$ and $y(n)$ are the state vector and the output of the nonlinear discrete time dynamical system, respectively, where $\mathbf{x}(n)$ is the binary sequence and the $k$th element of $\mathbf{x}(n)$ is $c_k$. Moreover, $y(n)$ is the corresponding

complex number. However, it is worth noting that $Y$ is not the whole complex plane. That means, some complex numbers cannot be represented by the binary combinations of the elements in $\Gamma$.

Since $\mathfrak{I}$ involves the modulo operator, the nonlinear discrete time dynamical system may exhibit both the fractal phenomenon and the chaotic phenomenon [22]. Figure A1 shows some computer numerical simulation results of $Y$ with $\angle\lambda = \frac{\pi}{10}$ for $|\lambda| = 1.1$, $|\lambda| = 1.3$, $|\lambda| = 1.5$ and $|\lambda| = 1.9$. On the other hand, Figure A2 shows some computer numerical simulation results of $Y$ with $|\lambda| = 1.5$ for $\angle\lambda = 0$, $\angle\lambda = \frac{\pi}{4}$, $\angle\lambda = \frac{\pi}{2}$ and $\angle\lambda = \pi$. It can be seen from Figure A1 that the size of $Y$ becomes smaller as $|\lambda|$ increases. This is because the distance between the complex numbers $\lambda^{-k}$ and $-\lambda^{-k}$ decreases as $|\lambda|$ increases. Besides, it is worth noting that $Y$ does not contain a neighborhood around the origin as $|\lambda|$ increases. In addition, if $\angle\lambda$ is an integer multiple of $\pi$, then $\lambda$ is a real number. In this case, $Y$ can only contain the real numbers in the range $\left(-\frac{\lambda}{\lambda-1}, \frac{\lambda}{\lambda-1}\right)$. If $\lambda$ is a real number with $|\lambda| \geq 2$, then $Y$ does not contain all real numbers in the range $\left(-\frac{\lambda-2}{\lambda-1}, \frac{\lambda-2}{\lambda-1}\right)$.

This investigation will be applied to derive a stability condition for the sigma delta modulator. Moreover, it will be used for formulating a frequency-domain-based bit flipping control strategy for stabilizing the sigma delta modulator.

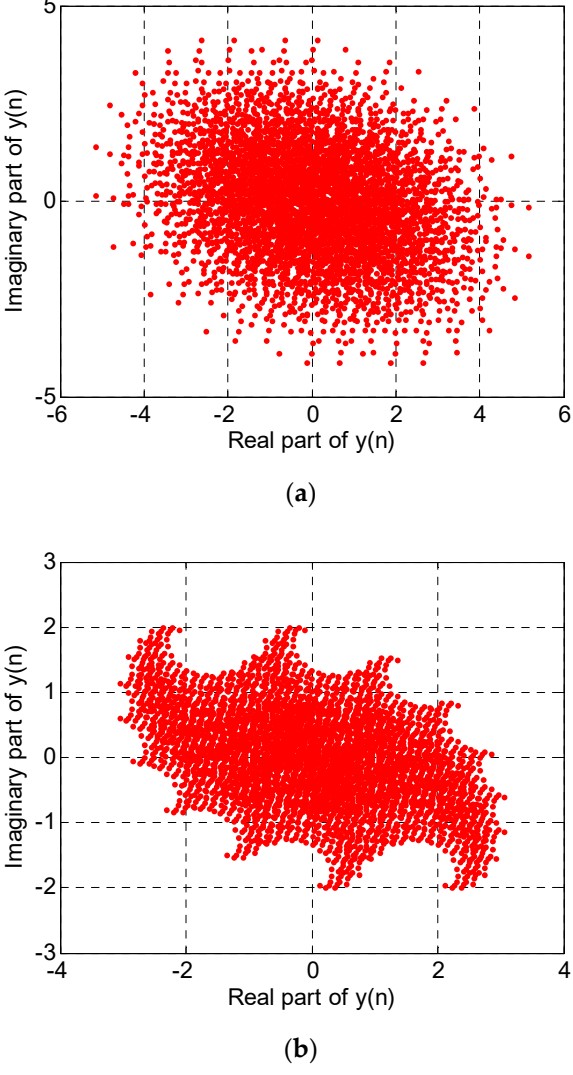

**(a)**

**(b)**

**Figure A1.** *Cont.*

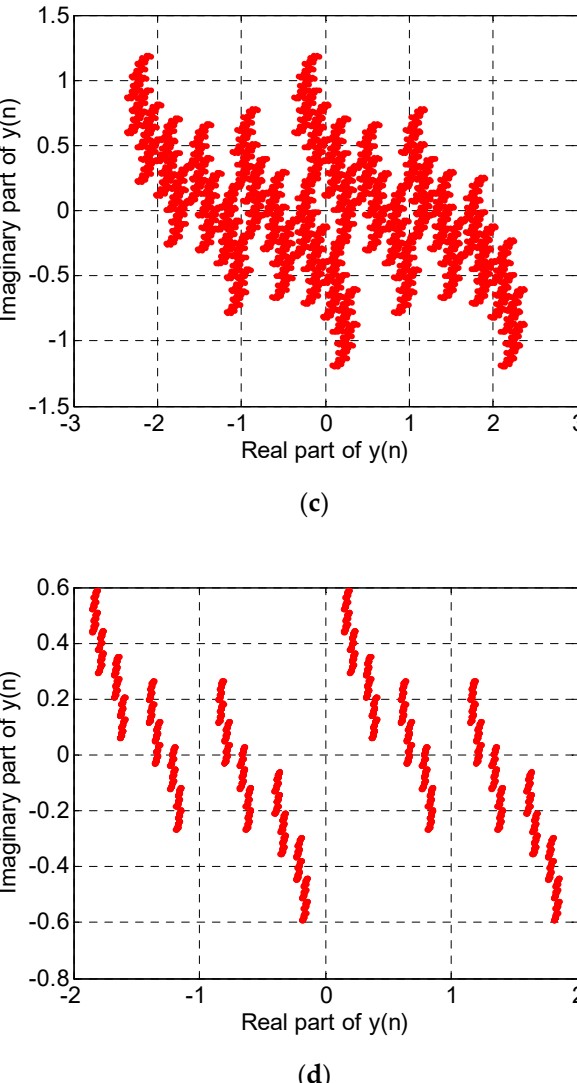

(**c**)

(**d**)

**Figure A1.** (**a**) $Y$ with $\angle\lambda = \frac{\pi}{10}$ and $|\lambda| = 1.1$. (**b**) $Y$ with $\angle\lambda = \frac{\pi}{10}$ and $|\lambda| = 1.3$. (**c**) $Y$ with $\angle\lambda = \frac{\pi}{10}$ and $|\lambda| = 1.5$. (**d**) $Y$ with $\angle\lambda = \frac{\pi}{10}$ and $|\lambda| = 1.9$.

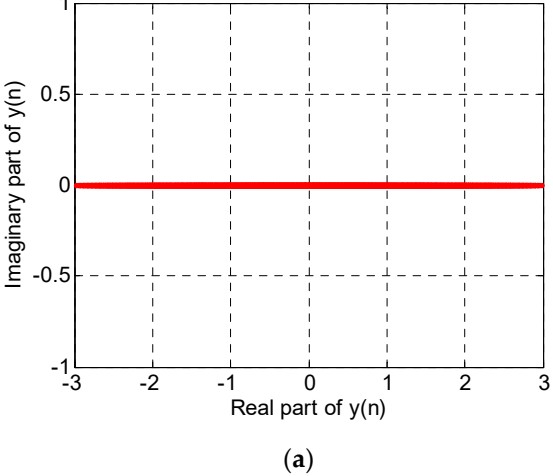

(**a**)

**Figure A2.** *Cont.*

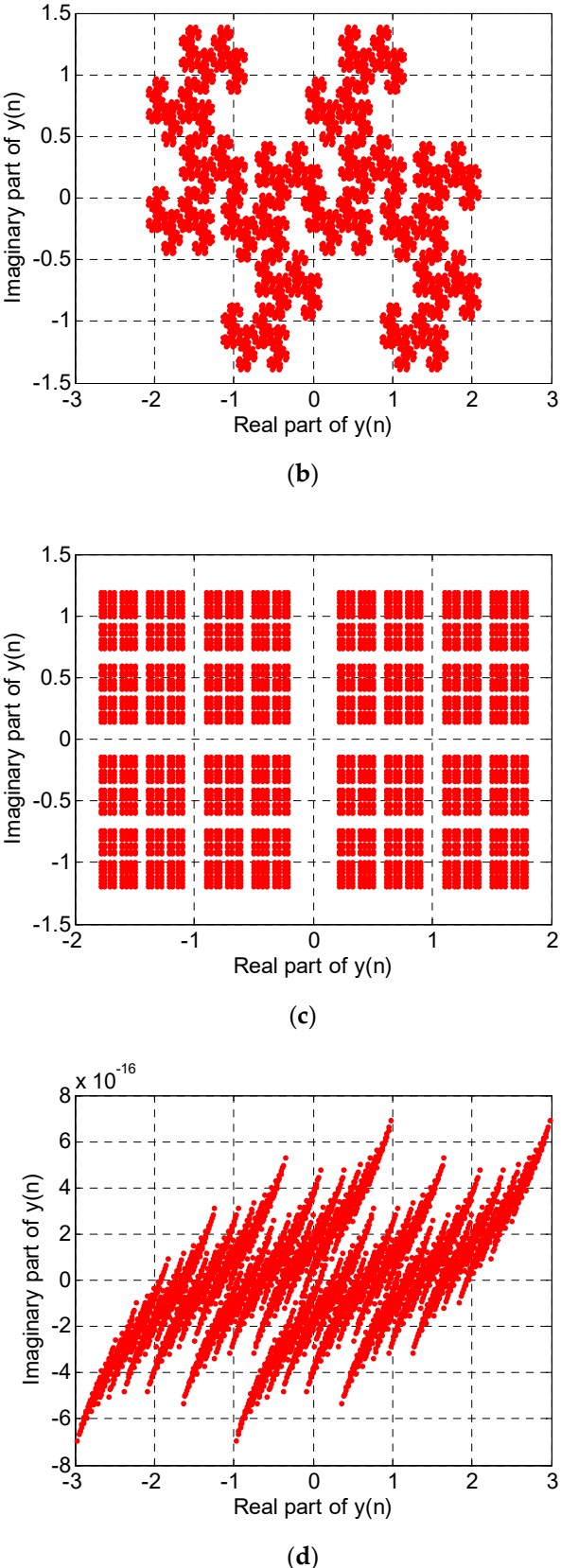

(**b**)

(**c**)

(**d**)

**Figure A2.** (**a**) $Y$ with $|\lambda| = 1.5$ and $\angle\lambda = 0$. (**b**) $Y$ with $|\lambda| = 1.5$ and $\angle\lambda = \frac{\pi}{4}$. (**c**) $Y$ with $|\lambda| = 1.5$ and $\angle\lambda = \frac{\pi}{2}$. (**d**) $Y$ with $|\lambda| = 1.5$ and $\angle\lambda = \pi$.

**Appendix C. Review on the Frequency-Domain-Based Bit Flipping Control Strategy**

Denote the transfer function of the loop filter as $F(z)$. Denote the noise transfer function and the signal transfer function as $NTF(z)$ and $STF(z)$, respectively. In order to achieve a high signal to noise ratio, it is desired to achieve $NTF(z) \to 0$ and $STF(z) \to 1$ in the signal band as well as $NTF(z) \to 1$ and $STF(z) \to 0$ in the noise band. Since $NTF(z) = \frac{1}{1+F(z)}$ and $STF(z) = \frac{F(z)}{1+F(z)}$, it is implied that $|F(z)| \to +\infty$ in the signal band and $F(z) = 0$ in the noise band. This further implies that $F(z)$ contains some unstable poles ((1) the poles outside the unit circle; or (2) the poles on the unit circle with the frequencies inside the signal band in which the resonance could occur). As the loop filter is unstable, the stability issue is very critical. As a result, the control of the sigma delta modulator is very important.

Assume that the loop filter could be realized via the state space representation with the state space matrices $\mathbf{A}$, $\mathbf{B}$, $\mathbf{C}$ and $D$. Here, $D = 0$. Let the input sequence and the quantized output sequence of the sigma delta modulator be $u(n)$ and $s(n)$, respectively. Denote the state vector as $\mathbf{x}(n)$. Then, it can be shown that:

$$\mathbf{x}(n) = \mathbf{A}^n \mathbf{x}(0) + \sum_{k=0}^{n-1} \mathbf{A}^{n-1-k} \mathbf{B}(u(k) - s(k))$$

Assume that $\mathbf{A}$ is diagonalizable. Denote a matrix $\mathbf{T}$ in such a way that the columns of $\mathbf{T}$ are the eigenvectors of $\mathbf{A}$. Moreover, denote a diagonal matrix $\mathbf{D}$ in such a way that its diagonal elements are the corresponding eigenvalues of $\mathbf{A}$. That is, $\mathbf{A} = \mathbf{T}\mathbf{D}\mathbf{T}^{-1}$. Let the diagonal elements of $\mathbf{D}$ be $d_i$. Assume that the state vectors are $N$ dimensional. Denote:

$$\tilde{\mathbf{x}}(n) = \begin{bmatrix} \widetilde{x}_0(n) & \cdots & \widetilde{x}_{N-1}(n) \end{bmatrix}^T = \mathbf{T}^{-1}\mathbf{x}(n)$$

and

$$\tilde{\mathbf{B}} = \begin{bmatrix} \widetilde{b}_0 & \cdots & \widetilde{b}_{N-1} \end{bmatrix}^T = \mathbf{T}^{-1}\mathbf{B}$$

Then, we have:

$$\mathbf{x}(n) = \mathbf{T}\mathbf{D}^n\mathbf{T}^{-1}\mathbf{x}(0) + \sum_{k=0}^{n-1} \mathbf{T}\mathbf{D}^{n-1-k}\mathbf{T}^{-1}\mathbf{B}(u(k) - s(k))$$

$$\tilde{\mathbf{x}}(n) = \mathbf{T}^{-1}\mathbf{x}(n) = \mathbf{D}^n\mathbf{T}^{-1}\mathbf{x}(0) + \sum_{k=0}^{n-1} \mathbf{D}^{n-1-k}\mathbf{T}^{-1}\mathbf{B}(u(k) - s(k))$$

$$= \mathbf{D}^n\tilde{\mathbf{x}}(0) + \sum_{k=0}^{n-1} \mathbf{D}^{n-1-k}\tilde{\mathbf{B}}(u(k) - s(k))$$

This implies that:

$$\widetilde{x}_i(n) = d_i^n \widetilde{x}_i(0) + \sum_{k=0}^{n-1} d_i^{n-1-k} \widetilde{b}_i(u(k) - s(k))$$

In other words, we have:

$$\frac{\widetilde{x}_i(n)}{\widetilde{b}_i d_i^{n-1}} = \frac{d_i \widetilde{x}_i(0)}{\widetilde{b}_i} + \sum_{k=0}^{n-1} \frac{u(k) - s(k)}{d_i^k}$$

Denote the z-transform of the input sequence and that of the quantized output sequence of the sigma delta modulator as $U(z)$ and $S(z)$, respectively. Suppose that the eigenvectors in $\mathbf{T}$ and the corresponding eigenvalues in $\mathbf{D}$ are arranged in such a way that $|d_i| \geq |d|_j$ for $j > i$. Moreover, assume that there are $P$ unstable eigenvalues in $\mathbf{D}$. That is, $|d_i| \geq 1$ for $i = 0, \cdots, P-1$. By flipping some

values in the quantizer output, denote the z-transform of the new quantizer output as $\hat{S}(z)$. In order to stabilize the sigma delta modulator, we have:

$$\lim_{n \to +\infty} \frac{\widetilde{x}_i(n)}{\widetilde{b}_i d_i^{n-1}} = 0 = \frac{d_i \widetilde{x}_i(0)}{\widetilde{b}_i} + U(z)\big|_{z=d_i} - \hat{S}(z)\big|_{z=d_i}$$

for $i = 0, \cdots, P-1$. This further implies that

$$\frac{d_i \widetilde{x}_i(0)}{\widetilde{b}_i} + U(z)\big|_{z=d_i} = \hat{S}(z)\big|_{z=d_i}$$

for $i = 0, \cdots, P-1$.

Define $Y_{d_i}$ as the set of complex numbers where the elements inside $Y_{d_i}$ can be represented as the binary combinations of the elements in the sequences $\{d_i^{-k} : k \geq 0\}$. That is, $\exists c_k \in \{1, -1\}\ \forall k \geq 0$ such that $\sum_{k \geq 0} c_k d_i^{-k} \in Y_{d_i}$. If $\frac{d_i \widetilde{x}_i(0)}{\widetilde{b}_i} + U(z)\big|_{z=d_i} \in Y_{d_i}$ for $i = 0, \cdots, P-1$, then there exists a binary sequence $c_k \in \{1, -1\}\ \forall k \geq 0$ such that the single bit interpolative sigma delta modulator is internally stable with the z-transform of the quantizer output equal to $\hat{S}(z)$.

The above result characterizes a condition for stabilizing the single-bit high-order interpolative sigma delta modulator via flipping some values of the quantizer output. However, it can be seen from Section 2 that $Y_{d_i}$ does not include all complex numbers. This implies that if $\frac{d_i \widetilde{x}_i(0)}{\widetilde{b}_i} + U(z)\big|_{z=d_i} \in C \backslash Y_{d_i}$, then there does not exist any binary sequence such that the sigma delta modulator can be stabilized. On the other hand, if $\frac{d_i \widetilde{x}_i(0)}{\widetilde{b}_i} + U(z)\big|_{z=d_i} \in Y_{d_i}$, then there exists a binary sequence such that the sigma delta modulator can be stabilized. For this case, it is required to find this binary sequence. In particular, an integer programming approach is employed to address this issue. The objective function of the optimization problem is to minimize the error energy between $\frac{d_i \widetilde{x}_i(0)}{\widetilde{b}_i} + U(z)\big|_{z=d_i}$ and $S(z)\big|_{z=d_i}$ subject to the binary constraint imposed to the quantizer output. That is, the optimization problem is formulated as follows:

**Problem (P)**

$$\min_{\mathbf{x}'} \sum_{\forall i \in \{0, \cdots, P-1\}} \left| \mathbf{\psi}_i^T \mathbf{x}' - \frac{d_i \widetilde{x}_i(0)}{\widetilde{b}_i} - U(z)\big|_{z=d_i} \right|^2,$$

subject to $x'_k \in \{1, -1\}\ \forall k \geq 0$.

Here, $x'_k$ is the $k^{\text{th}}$ element of $\mathbf{x}'$ and $d_i^{-k}$ is the $k^{\text{th}}$ element of $\mathbf{\psi}_i$. If there exists a set of binary sequences such that the single-bit high-order interpolative sigma delta modulator is internally stable, then the objective functional value of Problem (**P**) for these binary sequences in this set will be exactly equal to zero. In this case, $\mathbf{x}'$ is the global minimum of Problem (**P**). Hence, by finding a solution of Problem (**P**), we can find a stable binary sequence stabilizing the sigma delta modulator. However, it is difficult to find a solution of Problem (**P**). This is because it is an infinite dimensional optimization problem. To address this issue, the quantizer output bit streams are divided into blocks with the length of each block is equal to $M$. Denote the vector of the binary sequence in the $m^{\text{th}}$ block as $\mathbf{x}''_m$ for $m \geq 0$. Define a vector $\mathbf{\psi}''_{i,m}$ such that the $k^{\text{th}}$ element of $\mathbf{\psi}''_{i,m}$ is equal to $d_i^{-mM-k}$ for $k = 0, \cdots, M-1$. Denote $\mathbf{x}_m^*$ for $m \geq 0$ as a solution of the following finite dimensional integer programming problem:

**Problem (P$'_m$)**

$$\min_{\mathbf{x}'} \sum_{\forall i \in \{0, \cdots, P-1\}} \left| \mathbf{\psi}''_{i,m}{}^T \mathbf{x}''_m + \sum_{n=0}^{m-1} \mathbf{\psi}''_{i,n}{}^T \mathbf{x}_n^* - \frac{d_i \widetilde{x}_i(0)}{\widetilde{b}_i} - U(z)\big|_{z=d_i} \right|^2,$$

subject to

$$x''_{k,m} \in \{1, -1\} \quad \forall k \in \{0, \cdots, M-1\}.$$

here, $x''_{k,m}$ is the $k^{\text{th}}$ element of $\mathbf{x}''_m$. It is worth noting that Problem ($\mathbf{P}'_m$) is a standard integer programming problem. There are many existing algorithms [23] for finding a solution of Problem ($\mathbf{P}'_m$). Besides, it is worth noting that if $M$ is large enough, then the solution of Problem ($\mathbf{P}'_m$) will be close to that of Problem (**P**).

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
