# Peer review of "Implementation and Performance Evaluation of the Frequency-Domain-Based Bit Flipping Controller for Stabilizing the Single-Bit High-Order Interpolative Sigma Delta Modulators"

_applsci, doi:10.3390/app10175785_

Round 1

Reviewer 1 Report

The proposed approach in this paper for hardware implementation of the frequency domain-based bit flipping controller for stabilizing the single bit high order interpolative Sigma-Delta modulator previously published look very interesting and cost effective too. However, I have some comments:

1) A careful editing of the abstract is highly recommended to enhance your work and valorize your work outcome.

2) Professional English proofreading is required to correct typos error in many locations and also to improve the writing style.

3) Comparison to clipping method is ok but more discussion of the results is required. i.e. results shown in figure (4 to 7).

4) How much is the achieved SNR and how good it is compared to other methods.

Reviewer 2 Report

The paper says it is the hardware implementation of the system, however, there is very limited evidence of the hardware implementation apart from the brief section 4. The results are predominantly computer simulation based, therefore the claim of hardware implementation on the abstract is not warranted.

The paper organisation is poor and therefore difficult to follow. To increase the readability of the paper, it is recommended that the author clearly differentiate between the section that are literature review, background theory and the contributory section of the paper. Currently section 2 and 3 are very long and are mentioned as reviews, if they are reviews (which is not the scope of the paper), is it not a better idea to just briefly do a literature review to pin point the research gap and then make references to existing works.

The comparison of the work in simulation results with reference [4] needs justification and information needs to be provided on the selection criteria for the comparison. The paper is said to be extension to ref [20] so information needs to be provided what is the incremental contribution on this article. It is said in abstract that this is hardware implementation of [20], but as mentioned above, there is little evidence of hardware implementation and hardware results.

The paper needs to outline the significant novel contributions made, possibly on bullet points.

The reference of the paper needs to be sorted, currently it is not in order.

Round 2

Reviewer 2 Report

The authors have made the necessary changes and I am happy to accept this in its current form.